# Anaerobic Reactor Filling for Phosphorus Removal by Metal Dissolution Method

**DOI:** 10.3390/ma15062263

**Published:** 2022-03-18

**Authors:** Marcin Dębowski, Marcin Zieliński, Joanna Kazimierowicz

**Affiliations:** 1Department of Environmental Engineering, Faculty of Geoengineering, University of Warmia and Mazury in Olsztyn, 10-720 Olsztyn, Poland; marcin.zielinski@uwm.edu.pl; 2Department of Water Supply and Sewage Systems, Faculty of Civil Engineering and Environmental Sciences, Bialystok University of Technology, 15-351 Bialystok, Poland; j.kazimierowicz@pb.edu.pl

**Keywords:** active filling, anaerobic reactors, wastewater treatment, phosphorus removal, metal dissolution, microcellular extrusion

## Abstract

A commonly indicated drawback of anaerobic wastewater treatment is the low effectiveness of phosphorus removal. One possibility to eliminate this disadvantage is the implementation of active fillings that contain admixtures of metals, minerals, or other elements contributing to wastewater treatment intensification. The aim of the research was to present an active filling produced via microcellular extrusion technology, and to determine its properties and performance in anaerobic wastewater treatment. The influence of copper and iron admixtures on the properties of the obtained porous extrudate in terms of its functional properties was also examined. The Barus effect increased with the highest content of the blowing agent in the material from 110 ± 12 to 134 ± 14. The addition of metal powders caused an increase in the extrudate density. The modification of PVC resulted in the highest porosity, amounting to 47.0% ± 3.2%, and caused the tensile strength to decrease by about 50%. The determined values ranged from 211.8 ± 18.3 MPa to 97.1 ± 10.0 MPa. The use of the filling in anaerobic rectors promoted COD removal, intensified biogas production, and eliminated phosphorus with an efficiency of 64.4% to 90.7%, depending on the type of wastewater and applied technological parameters.

## 1. Introduction

Improving the effectiveness of removal of biogenic and organic compounds from wastewater is still the focus of interest of scientists, designers, and operators of municipal management facilities [1,2]. This interest is due to the growing ecological awareness of the population and to increasingly demanding quality standards set for wastewater discharged to receivers [3,4]. Due to the fact that simultaneous degradation of organic compounds and removal of phosphorus and nitrogen require complex and costly technologies, achieving the expected results is difficult in many cases and requires modernization [5]. These processes are based on providing many antagonistic technological parameters related to oxygen concentration, volatile fatty-acid content, and load or concentration of activated sludge [6,7]. In order to efficiently carry out the processes of organic compound degradation, orthophosphate binding, ammonification, nitrification, and denitrification, a typical technological system based on the use of variable anaerobic–aerobic conditions must in many cases be modified or supplemented with other elements affecting the treatment performance [8]. Therefore, there is a justified need to search for universal methods that will be competitive with the currently existing solutions in terms of investment costs and technology [9,10].

A growing interest has been observed in the last two decades in anaerobic technologies which are perceived as competitive with traditional aerobic methods [11]. Fermentation tanks offer a wide range of applications in highly loaded industrial and municipal wastewater treatment systems [12]. They are preferred due to their low energy consumption, lower sludge growth compared oxygen systems, limited spread of aerosols and odors, and quick start-up even after a long break in operation [13,14]. Another benefit is the production of biogas, which can be used for heating purposes or to generate electricity [15]. Efforts are still being made to improve the technological efficiency of anaerobic processes. Reactors with an innovative design have been used [16,17], physical factors have been implemented [18,19], and the pre-preparation of biodegradable pollutants [20] or the integration of anaerobic wastewater treatment with microalgae technologies has been increasingly introduced [21,22].

A commonly indicated drawback of anaerobic technologies is the low effectiveness of nitrogen and phosphorus removal [23]. The reduction in the concentration of biogenes in anaerobic processes is mainly due to the synthesis of the microbial biomass [24]. This drawback curbs the universality of this technology; therefore, its elimination is of fundamental importance to the further development of anaerobic methods [25]. One of the possibilities is the implementation of active fillings (AFs), the characteristics and properties of which intensify the processes of nutrient removal [26]. The characteristics of Afs should boost the removal effectiveness of organic substances, nutrients, and suspended solids, additionally eliminating or significantly reducing the shortcomings of conventional anaerobic methods [27]. The studies so far have focused only on presenting the technological efficiency related to the efficiency of pollutant removal and biogas production from wastewater of various origins, in rectors of various designs and operated under variable technological conditions. For the first time, detailed data characterizing the filling in terms of physical properties are presented, which is extremely important from the point of view of its implementation in practice and the assessment of the possibility of its use on a large scale.

AFs are usually elements made of plastic containing an admixture of metals, including mainly copper, aluminum, or iron, or other elements influencing the intensification of wastewater treatment [28]. This type of improvement usually allows for more efficient adaptation of reactors and faster achievement of the target technological performance [29]. The ecological effect of deploying AF is also associated with the reduction in secondary pollution of wastewater with chemical compounds that are used in the chemical precipitation of phosphorus with inorganic coagulants or polyelectrolytes [30]. Modified fillings can be produced from waste materials, which is in line with the assumptions of a circular economy [31]. Often, the fillings are universal in nature and can be used not only in anaerobic reactors, but also in biological beds, air-lift reactors, typical oxygen chambers, or constructed wetlands [32]. New types of fillings are still being sought that will feature a large active surface, high mechanical resistance, and resistance to degradation in the wastewater environment, will not lead to secondary pollution of the environment, can be regenerated or recycled, will allow for efficient improvement of wastewater treatment processes, and whose production will be profitable [33]. This publication presents novel, not commonly used methods allowing to increase the extent of wastewater treatment that can be introduced into the existing and functioning facilities. Identification of the applicability of the presented devices and technologies will allow for a real assessment of the feasibility of their introduction and application in specific wastewater treatment plants. It will also make it possible to reach producers and distributors of specific technologies.

The major goal of this manuscript is to present the production technology, properties, and technological performance of an active filling produced with the microcellular extrusion method dedicated for anaerobic reactors. The research includes a determination of the influence of copper and iron admixtures on the selected properties of the porous extrudate in terms of its functional properties meeting market demands. The technological performance of the characterized filling is also presented, using the example of studies carried out so far in reactors operated on a bench and fractional–technical scale. The studies so far have focused only on presenting the technological efficiency related to the efficiency of pollutant removal and biogas production from wastewater of various origins, in rectors of various designs and operated under variable technological conditions. For the first time, detailed data characterizing the filling in terms of physical properties are presented, which is extremely important from the point of view of its implementation in practice and the assessment of the possibility of its use on a large scale.

## 2. Materials and Methods

### 2.1. Scope of Research

In the experimental design, the direct output variables were as follows:− temperature distribution along the extrusion head, °C;− rotational speed of the screw, υ·s^−1^;− type of the cooling medium;− temperature (t) of the cooling medium, °C;− sample diameter (D_w_), mm;− extrusion head nozzle diameter (D_d_), mm;− sample length (L), mm.

The indirect output variables were as follows:− normal density (ρ_n_) of the sample, kg/m^3^;− porosity (SP) of the sample, %;− tensile strength (σ_r_) of the sample, MPa;− relative elongation (ε_r_) at break of the sample, %;− the Barus β effect, %.

The variable factors adopted were the content of blowing agent in the plastic (%), the iron content in the plastic (%), and the copper content in the plastic (%). The tested material was plasticized poly(vinyl chloride). The method of the blowing agent dosing and the structural elements of the plasticizing system, the container, the extrusion head, and other components of the microcellular extrusion technological line were considered as constant factors. There could also be some external factors in the research, such as the varying voltage of the electric current, air humidity, and ambient temperature. It is estimated that the impact of modifying these factors on the research results is very small and can be neglected without harm to the work. Table 1 presents the division of studies into variants and the numerical values of the variable factors.

### 2.2. Materials

Plasticized poly(vinyl chloride) (PVC), in the form of transparent granules, was used in the study. It was produced by Alfa Sp. z o.o. (Wroclaw, Poland) under the trade name Alfavinyl GFM/4—31—TR. The use of this type of poly(vinyl chloride) in the study was driven by its wide applicability and extensive processing with the extrusion method. Its basic properties were as follows: density 1230 kg/m^3^, modulus of elasticity 2600 MPa, tensile strength 21 MPa, elongation at break 300%, and Shore A hardness 80 °Sh. The PVC was modified with a granulated blowing agent Hydrocerol 530 (Clariant Masterbatch GmbH & Co. OHG, Ahrensburg, Germany) during processing. This agent has exothermic decomposition characteristics and is intended for the extrusion of porous polyvinyl chloride pipes in particular. A mixture containing 50% of a blowing and nucleidizing agent was used in the process. The initial degradation temperature of Hydrocerol 530 was 170 °C. Chemically pure copper and iron powders manufactured by Cometox were also mixed in with the PVC.

In order to select the appropriate mixture of preliminary substrates, which would enable constructing the active filling with catalytic properties, the material was mixed with chemically pure substances, i.e., copper and iron powders (Cometox SRL, San Giacomo, Italy). The amount of metal admixtures introduced in each case was 5.0% of the total mass of the filling. Depending on the experimental series, the weight ratio of copper to iron was 1_Cu_/9_Fe_ and 1_Cu_/4_Fe_. The content and ratios of metals were determined on the basis of the results of preliminary studies [34].

### 2.3. Experimental Station

The extrusion process of PVC modified with the blowing agent and metal powders was carried out on the laboratory stand of the pipe extrusion line. The technological line was equipped with a single-screw extruder with stepless adjustment of the screw rotation speed, an extrusion head, a cooling bath (18 ± 2 °C), and a collecting device. A straight pin extrusion head for pipes was used to test the extrusion process (Figure 1). This head has replaceable extrusion nozzles and is used to extrude closed and open circular profiles, such as mainly rods and pipes. The nozzle with a ring cross-section used in the experiment had an outer channel diameter of 19.5 mm and an inner channel diameter of 13.5 mm. The process was carried out using the T-32-25 extruder, intended for extrusion of thermoplastics in the form of granules (Figure 2).

The basic technical data of the extruder used in the experiment were as follows: capacity 3.0–12.0 kg/h, screw diameter 32 mm, screw length-to-diameter ratio 25, screw speed 0.55–2.00 s^−1^, number of heating zones of the cylinder 4, power of the drive motor 5.5 kW, power of heaters of the plasticizing system and the extrusion head 4 kW, and mass of the extruder 840 kg. The temperatures in the individual zones of the plasticizing system were as follows: hopper zone 110 °C, feeding zone 120 °C, conversion zone 140 °C, and metering zone 140 °C, while the temperature of the extrusion head was 160 °C. The microcellular extrusion process was initiated by stabilizing the distribution of the set temperature. Then, poly(vinyl chloride) was fed with the addition of a blowing agent, in accordance with the adopted feeding variants of the agent. A conventional screw designed for processing poly(vinyl chloride) was used in the study. Its construction diagram and basic dimensions are shown in Figure 3. The screw had a continuous screw channel over the entire length of the working part, as well as a unit reduction and a total reduction. It was made of tool steel and was nitrided and polished. The extrusion process was carried out at a rotational speed of the screw equal to 50 rpm. The process stabilized after 900 s.

### 2.4. Analysis of Mechanical Properties

The Barus effect, which is the value of the expansion of the material stream flowing from the die of the extrusion head, was determined from the following relationship:(1)$$\beta =\frac{D_{ZW}}{D_{ZD}}100\%,$$
where *D_ZW_* is the outer diameter of the pipe (mm), and *D_ZD_* is the outer diameter of the extrusion head nozzle (mm). The physical structure of the extrudate was tested on a stand equipped with a Vision DX-51 inspection microscope and an Olympus digital camera (E-620). Observations were made in reflected light. Photographs of the outer surface of the pipe and cross-section were taken.

Analyses of the selected physical properties of the obtained filling elements included measurements of density, porosity, tensile strength, and elongation at break. The apparent density of the obtained extrudate was tested in accordance with the PN-EN ISO 845:2000 standard. Analyses were made using samples in the form of cut-out fragments of the obtained pipes weighing from 1 to 5 g. The apparent density of the samples was calculated from the following formula:(2)$$\rho_{c}=\frac{m}{V},\mathrm{kg}/m^{3},$$
where *m* is the weighed sample mass (g), and *V* is the sample volume (cm^3^).

Porosity analyses determining the part of the total volume of the material per pores were conducted in accordance with the PN-EN ISO 845:2000 standard. The porosity of the samples was determined from the following formula:(3)$$p=(1-s)100=\frac{\rho -\rho_{c}}{\rho }100,\%,$$
where *s* is the tightness, *ρ* is the true density (kg/m^3^), and *ρ_c_* is the apparent density (kg/m^3^).

The analyses of mechanical properties included measurements of tensile strength and elongation at break. The samples in the shape of oars were cut out with a cutting die in accordance with the PN-EN ISO 527-3:1998 standard. The analyses were carried out on a Zwick/Roell Z2.5 testing machine following the PN-EN ISO 527-1:1998 and PN-EN ISO 1798:2001 standards. The tensile speed was assumed to be 50 mm/min and remained constant during testing. The test temperature was 23 °C, and the relative humidity was 50%. Hardness was determined using an indenter in accordance with the PN-EN ISO 2439:2009 standard.

### 2.5. Technological Effectivity of Active Filling

The technological efficiency of the AF is presented on the basis of the research carried out to date. It has been confirmed in studies on the treatment of wastewater of various origins and characteristics. The filling was tested in a multivariant manner, tested in reactors of various designs and with the use of various technological parameters, in laboratory conditions and on a semi-technical scale.

### 2.6. Statistical Analysis

The experimental variants were tested in triplicate. The statistical analysis of experimental results was conducted, and determination coefficients *R*^2^ were calculated using a STATISTICA 13.1 PL package (StatSoft, Cracow, Poland). One-way analysis of variance (ANOVA) was conducted to determine differences in groups. The HSD Tukey test was deployed to find significant differences between the analyzed variables. The results were considered significant at *p* = 0.05.

## 3. Results and Discussion

### 3.1. Filling Characteristics

The modification of PVC with the blowing agent and metal powders enabled obtaining a porous extrudate with a modified physical structure (Figure 4). Spherically shaped pores of various diameters were formed that could be observed in the cross-section and longitudinal section of the pipe (Figure 5). The metal powders added to the plastic settled on the outer and inner layer of the pipe (Figure 6). The modification with the blowing agent resulted in the modification of the PVC pipe structure from solid to porous only in the inner layer of the pipe, i.e., in its core (Figure 5). On the other hand, the addition of the filler in the form of copper and iron powders modified the outer surface of the pipe, but more metal powders still settled on the inner surface of the extrudate (Figure 6).

On the basis of the analyses of selected properties of the obtained filling elements, it can be concluded that the Barus effect increased with the increasing content of the blowing agent in the material. The change in the value of this parameter ranged from 110 ± 12 in V1 to 134 ± 14 in V12. The addition of metal powders caused no significant changes in the Barus effect (Table 2). Modification of the material with a blowing agent significantly reduced the extrudate density, thereby increasing its porosity (Table 2). The filling density in the variants without the blowing agent was above 1200 kg/m^3^. However, in the variants where the blowing agent was used at 1 wt.%, the filling density dropped below 700 kg/m^3^. The inclusion of metal powders in the preliminary substrate was found to increase the extrudate density (Table 2). The modification of PVC with the blowing agent resulted in the highest porosity, amounting to 47.0% ± 3.2%, recorded in V8. In V1 (PVC without modifiers), the porosity reached 0.0%, while introducing only metal powders into the processed PVC caused an increase in its values ranging from 3.9% ± 0.4% to 4.1% ± 0.5% (Table 2).

Correlations were observed between the amount of blowing agent introduced and the hardness of the filling elements; specifically, the value of hardness decreased with the increasing proportion of the blowing agent. In contrast, the addition of metal powders had no effect on the value of this parameter (Table 2). Similar correlations were found in the case of elongation at break, the values of which ranged from 160% ± 15% to 110% ± 12%.

As a result of the modification of poly(vinyl chloride), the tensile strength of the obtained extrudate also changed. The blowing agent addition at 0.5 wt.% caused the tensile strength to decrease by about 20%, while its addition at 1.0 wt.% reduced it by nearly 50%. The determined values ranged from 211.8 ± 18.3 MPa in V1 to 97.1 ± 10.0 MPa in V8 (Table 2). A greater decrease in tensile strength was observed with the addition of copper powder, whereas a slightly lesser one was observed with PVC modification using iron powder. The influence of the blowing agent and of the iron and copper content was demonstrated through very strong negative correlations in variants 1 to 4, 5 to 8, and 9 to 12 in the case of apparent density (*R*^2^ > 0.92) (Figure 7a), hardness (*R*^2^ > 0.95) (Figure 7c), and tensile strength (*R*^2^ > 0.96) (Figure 7d). Very strong positive correlations (*R*^2^ > 0.96) were found in the same variants for porosity (Figure 7b).

Various types of fillings currently used in wastewater treatment processes are mainly made of polymeric materials [35], the most common of which include low-density polyethylene (PE-LD) and high-density polyethylene (PE-HD), as well as polyvinyl chloride (PVC). These plastics are widely used due to their good mechanical, processing, and functional properties, as well as their availability on the market [36,37]. However, these materials need to be modified due to new avenues of application, the need to alter or improve selected properties, and the requirements imposed on such products. The modification of products in the extrusion process can be achieved by changing the technological conditions of the process and design features of the plasticizing system elements and processing tools, as well as by using supporting agents with a wide array of influence on the material being processed [37].

The modification involving the addition of the supporting agents takes place by introducing these materials during the processing of the material or directly during its production. These agents include physical modifiers improving mainly resilience, susceptibility to thermal molding, or formability [38]. In addition to popular supporting agents such as calcium carbonate, carbon black, graphite, silicon dioxide, molybdenum dioxide, mica, talc, wood flour, glass fibers, and natural fibers, new agents are also used as modifiers [39]. These include porous and microporous agents, also called blowing agents [40]. The gaseous products emitted during processing cause changes in the structure of the material, leading to a decrease in its polymeric material density, while maintaining its high rigidity, low thermal conductivity, and satisfactory mechanical resistance and elasticity modulus [41]. The growing interest in porous materials and the methods of their production has resulted in the emergence of a new processing method, namely, microcellular extrusion. In recent years, the microcellular extrusion of thermoplastics has been one of the most rapidly developing methods of processing these plastics. Its aim is to obtain mainly various microporous profiles and coatings of reduced density, free from hollows on the extrudate surface and showing minimal processing shrinkage while maintaining properties similar to those of the products extruded using the conventional method [42].

### 3.2. Technological Effectivity of Active Filling

The introduction of metal admixtures, mainly iron and aluminum, into the filling’s structure allows improving the removal effectiveness of phosphates, which intensifies the processes of eutrophication and degradation of natural water bodies. Electrochemical corrosion processes are applied to this end [43]. In the wastewater environment, corrosive metals represent a source of ions that bind orthophosphates to insoluble forms and are directly responsible for reducing phosphorus concentration in the treated wastewater [44]. It has been proven that, in addition to electrochemical corrosion, of great importance is also the biological corrosion induced by microorganisms colonizing the wastewater environment. Most bacteria are not directly involved in the corrosion process, but intensify its course by producing metabolites or assimilating those that stimulate this process [45].

The viability of active filling was tested in the anaerobic treatment of diary wastewater. The aim of this study was to determine the effects of magneto-active microporous packing media manufactured by extrusion technology and modified by the addition of relevant amounts of metal catalysts and magnetic activation on the effectiveness of simulated dairy wastewater treatment and biogas productivity in a pilot plant-scale hybrid anaerobic biofilm reactor with full mixing (MA-HABR). A previous study showed that the use of innovative fillers in the biofilm bed reactor increased the sorption of organic matter, biogas productivity, and the binding of biogenic compounds. The highest treatment effectiveness was obtained with a load of organic compounds ranging from 6.0 to 8.0 kg COD/m^3^·day. In these variants, COD removal effectiveness reached 80% and biogas productivity ranged from 256.7 to 310.9 dm^3^/kg of COD removed, whereas methane production ranged from 420.6 to 557.1 dm^3^/day. In addition, the process enabled removing from 82.9% to 90.7% of phosphorus compounds [46].

Another study investigated the feasibility of using the packing media in a vertical labyrinth-flow reactor. In particular, it aimed to analyze the impact of placing the active filling in various functional parts of the fermentation bioreactor (hydrolysis zone, methanogenesis zone) on the effectiveness of biodegradation of organic compounds, removal of biogenes, and biogas productivity. The study demonstrated a significantly positive effect of placing the active filling in the methanogenesis zone on the removal of organic matter and phosphorus. Approximately 86% COD removal effectiveness and efficient removal of suspended solids from wastewater were achieved. The phosphorus removal effectiveness ranged from 85% to 87%. The highest biogas production amounting to 337 dm^3^/day and biogas yield reaching 415 dm^3^/kg of COD removed were achieved upon the use of the magneto-active filling [47]. The purpose of another study was to determine the effect of the number of magneto-active filling (MAF) elements in the hydrolysis zone on the anaerobic treatment of model dairy wastewater. The study conducted on the pilot installation scale allowed achieving a high removal effectiveness of organic compounds (72–76%) and phosphorus (84–88%). A reduction in the concentration of nitrogen compounds and suspended solids in the effluent was also observed. The effectiveness of wastewater treatment was found to be directly related to the amount of MAF elements in the hydrolysis zone [48].

In another study, dairy wastewater was treated in one-stage anaerobic moving biofilm reactors (AMBRs) at ambient temperature. The bio-treatment was aided by abiotic processes (iron corrosion), owing to the incorporation of metallic iron into the microporous structure of the filling. The AMBR operated at the hydraulic retention time of 5.5 days and the organic load rate (OLR) of 3.9 and 7.5 kgCOD/m^3^·day. With the OLR of 3.9 kgCOD/m^3^·day, the COD removal was significantly higher in the variant with 2% iron in the filling (86.6% ± 7.8%) than in the variant without iron (81.3% ± 9.0%). The addition of Fe significantly enhanced phosphorus removal from about 50% in AMBR without Fe addition to 72.1% ± 8.4% in AMBR with 2% Fe. It was proven that acetoclastic methanogens predominated over hydrogenotrophic methanogens in the bioreactor, but the introduction of Fe to the filling mass significantly increased the population of Metanobacteriaceae [49].

The results presented in another study demonstrated the feasibility of the simultaneous use of active filling and microwave radiation in the process of dairy wastewater treatment. The most significant influence of active filling on the effectiveness of COD removal and the methane content in biogas was recorded at the initial loading of the batch reactor chamber ranging from 6.0 to 8.0 COD/dm^3^. In turn, a significant effect of microwave radiation was found only in the variant without the active filling in the reactor. The technological solutions used in the experiment had no significant effect on the effectiveness of nitrogen compound removal, while the phosphorus compound removal effectiveness exceeded 90%, regardless of the active filling composition [34].

Another study analyzed the technological effectiveness of sugar-industry effluent methane fermentation in a fluidized active filling reactor (FAF-R). In the OLR range of 4.0–6.0 kg COD/m^3^·day, the COD removal effectiveness exceeded 74%. It allowed obtaining the final concentration in the range from 879 ± 235 to 1141 ± 206 mg O_2_/dm^3^. In these experimental variants, the methane content in the biogas approximated 70%. Increasing the OLR to 6.0 kg COD/m^3^·day led to a significant reduction in the observed technological effects. The COD in the effluent increased and amounted to 2113 ± 255 mg O_2_/dm^3^ on average, with the removal effectiveness reaching 68.9% ± 4%. In addition, the methane content in the biogas dropped by 10%, i.e., to 61.9% ± 3.1%. The decrease in the effectiveness of wastewater treatment and methane fermentation was correlated with the observed decrease in pH to 6.75 ± 0.18 and the increase in the FOS/TAC ratio to 0.44 ± 0.2. The use of FAF-R was shown to improve the removal effectiveness of phosphorus compounds, which ranged from 64.4% ± 2.4% to 81.2% ± 8.2%, depending on the variant. Low concentrations of suspended solids in the effluent were also obtained [26].

The technological performance of the active filling in the processes of removing pollutants from wastewater under anaerobic conditions has been confirmed in experiments carried out on both a laboratory and a fractional–technical scale, as shown in Table 3.

### 3.3. Technological Effectivity of Other Used Fillings

Other authors used different kinds of fillings in anaerobic wastewater treatment technology. Huang et al. (2020) tested an anaerobic side-stream reactor (ASSR) filled with polyethylene carriers (15% fill) coupled with a membrane bioreactor. COD removal efficiency was 95.4%, and a reduction in TP concentration of 40.2% was achieved [50]. Gannoun et al. (2008) during the anaerobic purification of cheese whey wastewater in an upflow anaerobic filter (UAF) reactor filled with Flocor material achieved a COD removal efficiency from 72% ± 5% to 90.2% ± 0.5%, with the amount of biogas in the range from 0.47 ± 0.05 to 1.15 ± 0.28 dm^3^∙day at a methane content of 89 ± 0.3–89 ± 0.3 dm^3^_CH4_/kgCOD_removed_ [51]. Filling the polyurethane foam cubes of the concrete reactor contributed to the achievement of the COD removal efficiency at the level of 96% [52]. The use of the multi-fed anaerobic filter (MFR) by Puñal et al. (1999) allowed for a better biomass decomposition compared to the use of a single-fed reactor (SFR) [53]. The performance of anaerobic upflow fixed reactors for biomethanation of high-strength cheese whey using different support materials such as charcoal, gravel, brick pieces, PVC pieces, and pumice stones at 37 °C was studied. Among them, the charcoal fixed film reactor showed the best performance [54]. The results are summarized in Table 4. In anaerobic filters equipped with various types of fillings, the efficiency of phosphorus compound removal was not observed or determined. The main task of the filling was to create a surface for the growth of the anaerobic bacteria population and to limit their spread outside the reactor during wastewater treatment. The presented active filling, due to the foreignness of micropores, large active surface, and the ability to bind phosphorus, is a competitive solution in relation to the currently used methods.

## 4. Conclusions

A growing interest has been observed in the last two decades in anaerobic technologies which are perceived as competitive with conventional aerobic methods. Unfortunately, their drawback is the low effectiveness of removing biogenic compounds, including mainly phosphorus, which is perceived as the most important factor responsible for the eutrophication processes of natural aquatic ecosystems. This manuscript presented the production process, characteristics, and technological performance of an active filling made in the process of microcellular extrusion with the addition of metal powders.

Filling modification with a blowing agent was shown to modify the structure of the extrudate from solid to porous only in the inner layer of the filling. On the other hand, the addition of copper and iron powders to the filling modified the outer surface of the pipe, but more metal powders still settled on the inner surface of the extrudate. It was found that the Barus effect and porosity increased with the increasing content of the blowing agent in the material, while the addition of metal powders caused no changes in their values. The Barus effect increased from 110 ± 12 to 134 ± 14, with the porosity amounting to 47.0% ± 3.2% and the tensile strength decreasing by about 50%. Filling modification with the blowing agent decreased the extrudate density and tensile strength by about 50%. A greater decrease in tensile strength was observed with the addition of copper powder to 97.1 ± 10.0 MPa, compared to PVC modification using iron powder. The use of filling in anaerobic rectors promoted COD removal, intensified biogas production, and eliminated phosphorus with an efficiency of 64.4% to 90.7%, depending on the type of wastewater and applied technological parameters.

## Figures and Tables

**Figure 1 materials-15-02263-f001:**
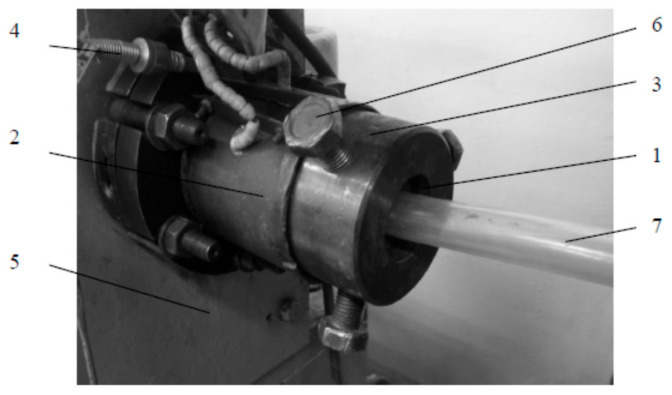
Extrusion head used in the study: 1—head nozzle body, 2—ring heater, 3—head body, 4—temperature sensor, 5—extruder casing fragment, 6—nozzle body adjustment screws, 7—filling produced.

**Figure 2 materials-15-02263-f002:**
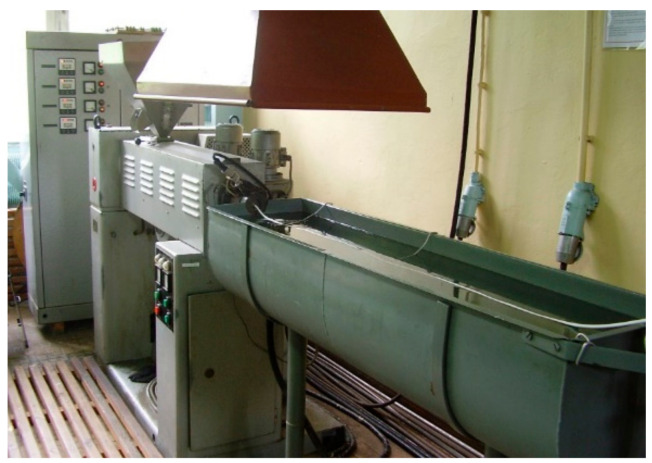
Station for testing the process of microcellular extrusion.

**Figure 3 materials-15-02263-f003:**
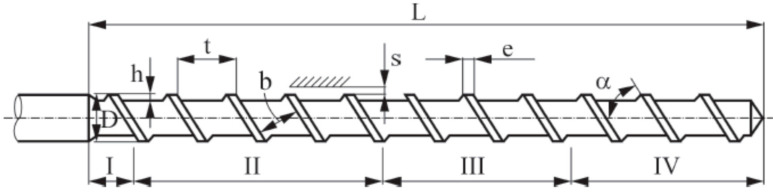
Construction diagram of the screw: I—hopper zone, II—feeding zone, III—conversion zone, IV—metering zones; L = 800 mm, D = 32 mm, h = 4 mm, t = 45 mm, e = 5 mm, s = 0.9 mm (the distance between the outer surface of the coil and the inner surface of the extruder barrel), α = 60°.

**Figure 4 materials-15-02263-f004:**
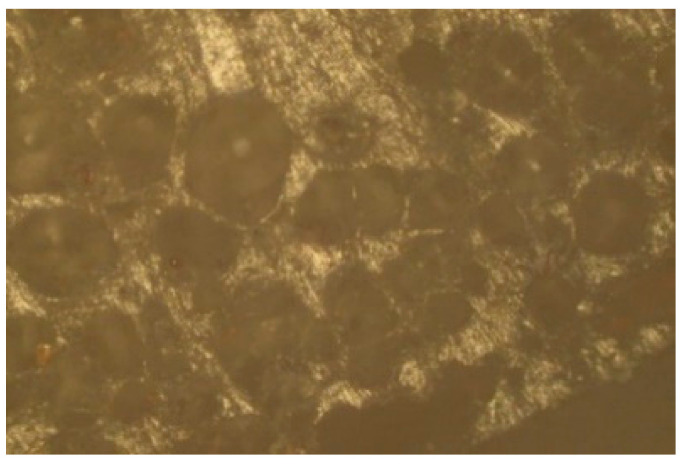
Element of the active filling.

**Figure 5 materials-15-02263-f005:**
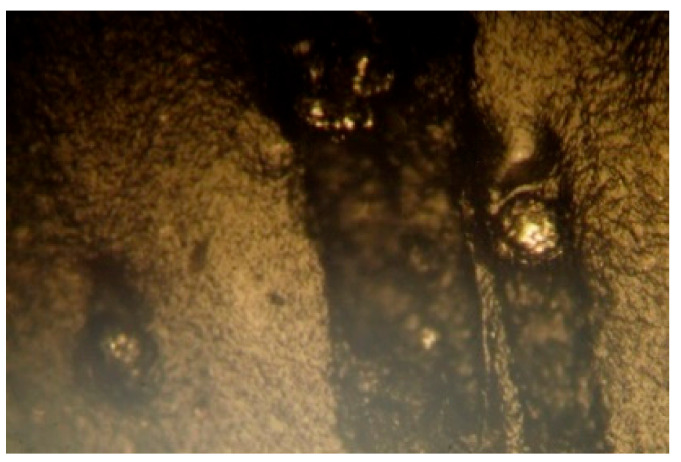
A fragment of a porous pipe in longitudinal section with visible pores.

**Figure 6 materials-15-02263-f006:**
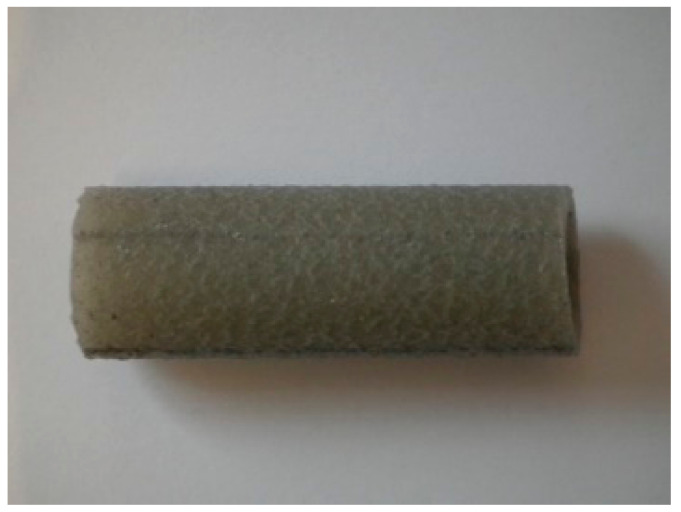
A fragment of the outer layer of a porous pipe with visible metal grains.

**Figure 7 materials-15-02263-f007:**
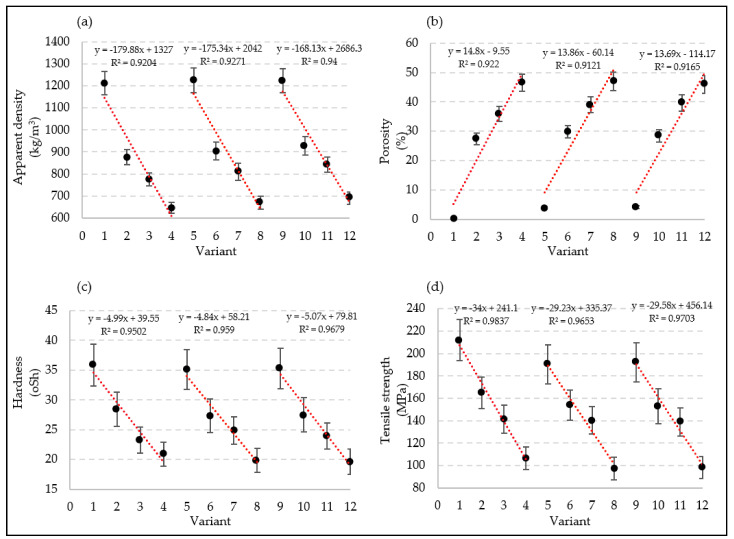
Changes in mechanical properties and correlations depending on the used variant of the filling modification: (**a**) apparent density; (**b**) porosity; (**c**) hardness; (**d**) tensile strength.

**Table 1 materials-15-02263-t001:** Numerical values of the variable factors.

Variant (V)	Blowing Agent Content, wt.%	Iron Content, wt.%	Copper Content, wt.%
1	0	0	0
2	0.5	0	0
3	0.8	0	0
4	1.0	0	0
5	0	4.5	0.5
6	0.5	4.5	0.5
7	0.8	4.5	0.5
8	1.0	4.5	0.5
9	0	4.0	1.0
10	0.5	4.0	1.0
11	0.8	4.0	1.0
12	1.0	4.0	1.0

**Table 2 materials-15-02263-t002:** Values of the analyzed characteristics of the filling.

Variant	Barus Effect *β* %	Apparent Density *ρ_c_*, kg/m^3^	Porosity *p*, %	Hardness, °Sh	Tensile Strength *σ_r_*, MPa	Elongation at Break *ε_r_*, %
1	110 ± 12	1211.6 ± 52.6	0.0 ± 0.0	35.8 ± 3.5	211.8 ± 18.3	160 ± 15
2	120 ± 11	876.9 ± 35.5	27.4 ± 2.0	28.4 ± 2.9	164.9 ± 14.2	140 ± 14
3	129 ± 10	774.3 ± 29.4	35.9 ± 2.6	23.2 ± 2.2	141.4 ± 12.5	120 ± 12
4	111 ± 12	646.2 ± 25.1	46.5 ± 2.9	20.9 ± 2.0	106.3 ± 10.1	110 ± 10
5	123 ± 13	1224.6 ± 55.8	3.9 ± 0.4	35.1 ± 3.3	190.1 ± 17.4	140 ± 15
6	131 ± 14	903.2 ± 40.5	29.8 ± 2.1	27.3 ± 2.8	153.8 ± 13.5	120 ± 13
7	130 ± 14	810.0 ± 38.6	39.1 ± 2.7	24.8 ± 2.3	140.5 ± 12.2	110 ± 12
8	110 ± 11	671.2 ± 29.5	47.0 ± 3.2	19.8 ± 2.0	97.1 ± 10.0	110 ± 11
9	121 ± 12	1223.0 ± 54.2	4.1 ± 0.5	35.3 ± 3.4	192.1 ± 17.5	150 ± 16
10	130 ± 14	927.4 ± 42.2	28.5 ± 2.1	27.5 ± 2.9	153.0 ± 15.5	120 ± 12
11	131 ± 13	842.7 ± 33.5	39.7 ± 2.8	23.9 ± 2.2	138.9 ± 12.8	120 ± 13
12	134 ± 14	690.8 ± 27.6	46.0 ± 3.0	19.6 ± 2.1	98.2 ± 10.0	110 ± 10

**Table 3 materials-15-02263-t003:** Summary of research works verifying the technological performance of an active filling in anaerobic reactors.

Reactor Type	Wastewater Type	COD Removal Effectiveness, %	Phosphorus Removal Effectiveness, %	Biogas Yield, dm^3^/kg CODrem.	Content of Methane in Biogas, %	Ref.
Fluidized active filling reactor (FAF-R)	Sugar-industry effluent	>74	64.4 ± 2.4–81.2 ± 8.2	356 ± 25–427 ± 14	70	[26]
Anaerobic reactor with active filling (AF), heated with microwave (EMR)	Dairy wastewater	88	>90	380 ± 17	64	[34]
Magneto-active hybrid anaerobic biofilm reactor (MA-HABR)	Dairy wastewater	80	82.9–90.7	256.7–310.9	61–68	[46]
Vertical reactor with labyrinth flow	Dairy wastewater	86	85–87	415	66–67	[47]
Vertical reactor with labyrinth flow	Dairy wastewater	72–76	84–88	-	-	[48]
Anaerobic moving biofilm reactors (AMBRs)	Dairy wastewater	86.6 ± 7.8	72.1 ± 8.4	130.0 ± 56.1	25.8–83.8	[49]

**Table 4 materials-15-02263-t004:** Technological efficiency of other fillings used in the processes of anaerobic wastewater treatment.

Reactor Type		Wastewater Type	COD Removal Effectiveness, %	Phosphorus Removal Effectiveness, %	Biogas Yield	Content of Methane in Biogas	Ref.
Anaerobic side-stream reactor (ASSR) filled with polyethylene carriers coupled with membrane bioreactor (MBR) filling fraction of 15%		Wastewater from the grit chamber	95.4	40.2	-	-	[50]
Upflow anaerobic filter (UAF) reactor filled with Flocor (φ3L3, porosity 95%, active surface 230 m^2^·m^−3^)		Cheese whey wastewater	72 ± 5–90.2 ± 0.5	-	0.47 ± 0.05–1.15 ± 0.28 dm^3^·day	89 ± 0.3–89 ± 0.3 dm^3^_CH4_/kg COD removed	[51]
Anaerobic batch reactor filled with polyurethane foam cubes with 0.5 cm sides and an apparent density of 23 kg/m^3^		Dairy whey wastewater	96	-	-	-	[52]
UAF reactors filled withraschig rings of corrugated polyvinyl chloride (PVC)	Multi-fed anaerobic filter (MFR)	Wastewaters from a tuna-processing factory	87	-	-	-	[53]
	Single-fed reactor (SFR)		65				
Anaerobic upflow fixed film reactors filled with:	Charcoal	Cheese whey wastewater	76.6	-	6.0 ± 0.011 dm^3^/day/dm^3^ digester	70% ± 0.66%	[54]
	Brick pieces		71.9		5.1 ± 0.020 dm^3^/day/dm^3^ digester	68% ± 0.83%	
	Gravel		73.7		5.4 ± 0.008 dm^3^/day/dm^3^ digester	68% ± 1.33%	
	PVC pieces		69.7		4.3 ± 0.013 dm^3^/day/dm^3^ digester	67% ± 0.16%	
	Pumice stone		67.5		3.8 ± 0.013 dm^3^/day/dm^3^ digester	64% ± 0.83%	

## Data Availability

Not applicable.

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
