# Peer review of "Anaerobic Reactor Filling for Phosphorus Removal by Metal Dissolution Method"

_materials, 2022, doi:10.3390/ma15062263_

Round 1
Reviewer 1 Report
In this work, an active filling was prepared with a goal to application in anaerobic reactors and the properties were investigated. However, the real application of this material is missing and the discussion of the materials was weak. The depth data analysis and the performance of the materials were needed for revision, the current version was not suitable for publication.
- Abstract: no result can be found in abstract, and the description of research background occupy large portion of this part.
- Introduction: this part was too long.
- Materials and methods: the application of the prepared materials was needed.
- The data was presented in a too simple way in the text, e.g. some figures can be merged as one; and some other figures can be added to make a depth discussion of the material and its performance.
- Section 3.2. this part looks like a review work, which is not suitable to be presented in a research-type manuscript.
- What is the performance of the materials prepared in this work?
Author Response
Detailed response to remarks of Reviewer #1:
Reviewer comment 1:
In this work, an active filling was prepared with a goal to application in anaerobic reactors and the properties were investigated. However, the real application of this material is missing and the discussion of the materials was weak. The depth data analysis and the performance of the materials were needed for revision, the current version was not suitable for publication.
Response:
Authors are grateful for all remarks and suggestions made by Reviewer regarding the manuscript entitled: “Anaerobic reactors filling for phosphorus removal by metal dissolution method”. (ID materials-1617807). All of them were considered in the revised manuscript. In addition, the manuscript was corrected in terms of editorial standards as well as brevity of English language.
The authors do not fully share the Reviewer's opinion on the poor scientific discussion and low-quality documentation of the application possibilities of the presented filling of anaerobic reactors. The authors hope that the answers to the Reviewer's questions and a detailed explanation of any doubts will be satisfactory and that the manuscript will be approved for publication. With kind regards, Authors
Below find, please, a detailed response to Reviewer comments.
Reviewer comment 2:
Abstract: no result can be found in abstract, and the description of research background occupy large portion of this part.
Response:
The authors are grateful for these remarks and agree with the Reviewer's opinion. The abstract has been corrected and supplemented as suggested in the review. The new version of Abstract is as follows:
„A commonly indicated drawback of anaerobic wastewater treatment is the low effectiveness of phosphorus removal. The possibility to eliminate this disadvantage is the implementation of active fillings that contain admixtures of metals, minerals or other elements contributing to wastewater treatment intensification. The aim of the research was to present the active filling produced via microcellular extrusion technology, and to determine its properties and performance in the anaerobic wastewater treatment. The influence of copper and iron admixtures on the properties of the obtained porous extrudate in terms of its functional properties was examined as well. Barus effect increases with the highest content of the blowing agent in the material from 110±12 to 134±14. The addition of metal powders caused increase the extrudate density. The modification of PVC resulted in the highest porosity, amounting to 47.0±3.2% and caused the tensile strength to decrease by about 50%. The determined values ranged from 211.8±18.3 MPa to 97.1±10.0 MPa. The use of filling in anaerobic rectors promoted COD removal, intensified biogas production and eliminated phosphorus with an efficiency of 64.4% to 90.7%, depending on the type of wastewater and applied technological parameters.”
Reviewer comment 3:
Introduction: this part was too long.
Response:
The authors thank the Reviewer for this critical remark. It was included in the modified version of the manuscript. From the Introduction section, subchapter: "Technical and technological bases" has been removed, modified and placed in the Discussion. Please check with the revised version of the manuscript.
Reviewer comment 4:
Materials and methods: the application of the prepared materials was needed.
Response:
Thanks a lot for this comment. The authors probably do not fully understand the Reviewer's intentions. Nevertheless, the authors will try to explain the inspiration behind writing the manuscript and the idea of presenting the results. The material and methods section presents in detail the characteristics of the production technology of active fill (AF) elements as well as analytical methods for testing its features and strength and mechanical properties. According to the authors, this approach to research and its presentation fully corresponds to the profile of the Materials journal. The authors focused on the production of AF and the presentation of its properties.
The technological efficiency of the use of AF has been confirmed in studies on the treatment of wastewater of various origins and characteristics. The filling was tested in a multivariant manner, tested in reactors of various designs and with the use of various technological parameters, in laboratory conditions and on a semi-technical scale. This has been presented and summarized in chapter 3.2. Technological effectivity of active filling. In the opinion of the authors, presenting further technological research is not justified since the effectiveness of the solution has already been confirmed.
To methodology section new chapter was added:
2.5. Technological effectivity of active filling
The technological efficiency of the AF was presented based on the research carried out to date. It has been confirmed in studies on the treatment of wastewater of various origins and characteristics. The filling was tested in a multivariant manner, tested in reactors of various designs and with the use of various technological parameters, in laboratory conditions and on a semi-technical scale.
Reviewer comment 5:
The data was presented in a too simple way in the text, e.g. some figures can be merged as one; and some other figures can be added to make a depth discussion of the material and its performance.
Response:
In line with the Reviewer's comment, the manuscript has been supplemented with figures showing changes in the mechanical properties of the AF depending on the production variants used. The figures show the correlations between the way of modifying the filling and the values describing mechanical and strength parameters. Please see modified version of manuscript.
Reviewer comment 6:
Section 3.2. this part looks like a review work, which is not suitable to be presented in a research-type manuscript.
Response:
Authors are grateful the Reviewer for this critical remark. This was partially explained in response to comment 4. The research part of the manuscript concerned the methodology for the production of the AF and the study of its strength and mechanical properties. It was described in accordance with the standards of research work, in terms of methodology and presentation of results. On the other hand, the confirmation of the technological effectiveness (anaerobic wastewater treatment efficiency) of the AF was presented on the basis of the research and implementation works carried out so far. In the opinion of the authors, it was not justified to perform further technological tests. This approach to the manuscript fully corresponds to the profile of journal Materials. It is not a journal dealing with technological issues, but rather concentrates on the material side of solutions.
In the revised version of the manuscript, chapter 3.2 is entitled Technological effectivity of active filling. Chapter 3.3.Technological effectivity of other used filling has been added, where the effectiveness of other anaerobic reactor fillings that have been tested so far has been discussed and summarized in a table 4. Technological efficiency of other fillings used in the processes of anaerobic wastewater treatment.
Please see modified version of manuscript.
Reviewer comment 7:
What is the performance of the materials prepared in this work?
Response:
Technological performance of filling active in terms of the removal of organic compounds, phosphorus, biogas production and methane content from wastewater in anaerobic treatment processes has been described and additionally discussed (agree with other Reviewers comments) in chapter 3.2. Technological effectivity of active filling. The most important data concerning technological performance are presented in table 3.
Reviewer 2 Report
This manuscript investigated production process, characteristics and technological performance of an active filling made in the process of microcellular extrusion with the addition of metal powders, which yielded some possible application for wastewater treatment. The work is interesting, however, there were some major concerns should be addressed before its publication.
- Line 51, in the anaerobic process, why is the effectiveness of nitrogen removal low? Kindly discuss some paper ( Total Environ. ,2022, 807, 150975). Moreover, AF seems cannot improve nitrogen removal.
- In the materials and methods part, the anaerobic reactor and the operating parameters should be introduced.
- Section 3.2, the performance of MA-HAMBR in terms of COD, biogas production should be given in figures.
- Line 318, same as above, kindly show the results in figures.
- I think this is a research manuscript, however, the performance results all shown in table 3 is more like a review paper. So kindly suggest the authors to present the results in a more appropriate way.
Author Response
Detailed response to remarks of Reviewer #2:
Reviewer comment 1:
This manuscript investigated production process, characteristics and technological performance of an active filling made in the process of microcellular extrusion with the addition of metal powders, which yielded some possible application for wastewater treatment. The work is interesting, however, there were some major concerns should be addressed before its publication.
Response:
Authors are grateful for all remarks and suggestions made by Reviewer regarding the manuscript entitled: “Anaerobic reactors filling for phosphorus removal by metal dissolution method”. (ID materials-1617807). All of them were considered in the revised manuscript. In addition, the manuscript was corrected in terms of editorial standards as well as brevity of English language. With kind regards, Authors.
Below find, please, a detailed response to Reviewer comments.
Reviewer comment 2:
Line 51, in the anaerobic process, why is the effectiveness of nitrogen removal low? Kindly discuss some paper ( Total Environ. ,2022, 807, 150975). Moreover, AF seems cannot improve nitrogen removal.
Response:
The efficiency of nitrogen removal in typical methane fermentation reactors for wastewater treatment is extremely low and results only from the synthesis of biomass of anaerobic microorganisms. Hundreds of studies published so far confirm this fact. Paper (Total Environ., 2022, 807, 150975) deals with a completely different technology, namely Anammox (anoxic ammonium oxidation). It is a technology that allows the reduction of nitrogen concentration in wastewater with limited oxygen availability. The Anammox process is based on the transformation of ammonia into gaseous nitrogen in an anaerobic environment, but the condition for the proper course of this reaction is the presence of azotine, which are electron acceptors, so the initiation of the Anammox process requires a prior nitrification stage (oxygen stage). The authors emphasize that we are talking about a completely different technology of wastewater treatment.
It is true that the AF presented in the manuscript does not improve the efficiency of nitrogen removal, but it is possible to develop AF in this direction. The authors are currently testing zeolite-enriched AF (minerals that can fix ammonium nitrogen). We are at the stage of developing the filling regeneration technology after the sorption capacity is fully exhausted.
Reviewer comment 3:
In the materials and methods part, the anaerobic reactor and the operating parameters should be introduced.
Response:
The material and methods section presents in detail the characteristics of the production technology of AF elements as well as analytical methods for testing its features and strength and mechanical properties. According to the authors, this approach to research and its presentation fully corresponds to the profile of the Materials journal. The authors focused on the production of AF and the presentation of its properties.
The technological efficiency of the use of AF has been already confirmed in studies on the treatment of wastewater of various origins and characteristics. The filling was tested in a multivariate manner, tested in reactors of various designs and with the use of various technological parameters, in laboratory conditions and on a semi-technical scale. This has been presented and summarized in chapter 3.2. Technological effectivity of active filling. In the opinion of the authors, presenting further technological research is not justified since the effectiveness of the solution has already been confirmed.
To methodology section new chapter was added:
2.5. Technological effectivity of active filling
The technological efficiency of the AF was presented based on the research carried out to date. It has been confirmed in studies on the treatment of wastewater of various origins and characteristics. The filling was tested in a multivariant manner, tested in reactors of various designs and with the use of various technological parameters, in laboratory conditions and on a semi-technical scale.
Reviewer comment 4:
Section 3.2, the performance of MA-HAMBR in terms of COD, biogas production should be given in figures. Line 318, same as above, kindly show the results in figures.
Response:
The effectiveness of the MA-HAMBR reactor is presented in the manuscript: "The highest treatment effectiveness was obtained with a load of organic compounds ranging from 6.0 to 8.0 kg COD/m3·d. In these variants, COD removal effectiveness reached 80%, biogas productivity ranged from 256.7 to 310.9 dm3 kg of COD removed, whereas methane production from 420.6 to 557.1 dm3/d. In addition, the process enabled removing from 82.9 to 90.7% of phosphorus compounds [40]” and also in Table 3. Summary of research works verifying the technological performance of an active filling in anaerobic reactors. According to the authors, the multiplication of the same results should be avoided. On the other hand, the interested reader may refer to the source publications for details:
Dębowski, M.; Zieliński, M.; Kisielewska, M.; Krzemieniewski, M.; Makowska, M.; Grądkowski, M.; Tor-Świątek, A. Simulated dairy wastewater treatment in a pilot plant scale magneto-active hybrid anaerobic biofilm reactor (MA-HABR). Braz. J. Chem. Eng. 2018, 35, 553–562. https://doi.org/10.1590/0104-6632.20180352s20170036
Kisielewska, M.; Dębowski, M.; Zieliński, M.; Krzemieniewski, M. Enhancement of Dairy Wastewater Treatment in a Combined Anaerobic Baffled and Biofilm Reactor with Magneto-Active Packing Media. J. Ecol. Eng. 2018, 19, 165–171. https://doi.org/10.12911/22998993/89816.
Reviewer comment 5:
I think this is a research manuscript, however, the performance results all shown in table 3 is more like a review paper. So kindly suggest the authors to present the results in a more appropriate way.
Response:
We thank the Reviewer for this critical remark. This was partially explained in response to comment 3. The research part of the manuscript concerned the methodology for the production of the active filling and the study of its strength and mechanical properties. It was described in accordance with the standards of research work, in terms of methodology and presentation of results. On the other hand, the confirmation of the technological effectiveness of the active filling was presented on the basis of the research and implementation works carried out so far. In the opinion of the authors, it was not justified to perform further technological tests. This approach to the issue fully corresponds to the profile of journal Materials. It is not a journal dealing with technological issues of wastewater treatment processes, but rather concentrates on the material side of solutions.
In the revised version of the manuscript, chapter 3.2 is entitled Technological effectivity of active filling. Chapter 3.3.Technological effectivity of other used filling has been added, where the effectiveness of other anaerobic reactor fillings that have been tested so far has been discussed and summarized in a table 4. Technological efficiency of other fillings used in the processes of anaerobic wastewater treatment.
Please see modified version of manuscript.
Reviewer 3 Report
The title is clear.
The content is in accord with title.
The manuscript adheres to the journal's standards after major revision.
The size of the article is appropriate to the contents.
The authors must underline the major findings of their work and explain novelty of this study comparatively with their published papers or other similar studies.
The Abstract must be revised. The Abstract section refers to the study findings, methodologies, discussion as well as conclusion.
The key words permit found article in the current registers or indexes.
In the introduction it is clearly described the state of the art of the investigated problem.
Technical and technological bases… is not very clear… is as Introduction part?
The 2.1. Scope of research
In the experimental design, the direct output variables included: temperature 144 distribution along the extrusion head, ºC; rotational speed of the screw, υ s-1;……
The enumeration is hard to watch.
The methods are well described and the equipment and materials have been adequately described.
The paper was written in standard, grammatically correct English, small corrections are necessary.
The figures have a good quality.
Please verify all figures.
The tables contain necessary results.
Please provide comparison with other studies. It is necessary, in tabular form, to provide the comparison studies.
The Conclusion must be revised. The main results must be presented in this section.
Please present full form of some. abbreviations when they first appeared. For example Afs, row 71.
The references aren’t in journal’s format. Please respect guide for authors.
For example:
Ashraf, A.; Ramamurthy, R.; Rene, E. R. Wastewater treatment and resource recovery technologies in the brewery industry: 449 Current trends and emerging practices. Sustainable Energy Technologies and Assessments 2021, 47, 450 101432
Journal wasn’t abbreviated.
There are journals with or without abbreviation, etc.
If the paper is in Materials journal topics, please proved 2 references from this journal (last year).
The paper is relatively easy to understand by readers from other area.

Author Response
Detailed response to remarks of Reviewer #3:
Reviewer comment 1 (all manuscript strengths and review summary):
The title is clear. The content is in accord with title. The size of the article is appropriate to the contents. The key words permit found article in the current registers or indexes. In the introduction it is clearly described the state of the art of the investigated problem. The methods are well described and the equipment and materials have been adequately described. The paper was written in standard, grammatically correct English, small corrections are necessary. The figures have a good quality. The tables contain necessary results. The paper is relatively easy to understand by readers from other area. The manuscript adheres to the journal's standards after major revision.
Response:
Authors are grateful for all remarks and suggestions made by Reviewer regarding the manuscript entitled: “Anaerobic reactors filling for phosphorus removal by metal dissolution method”. (ID materials-1617807). All of them were considered in the revised manuscript. In addition, the manuscript was corrected in terms of editorial standards as well as brevity of English language. With kind regards, Authors.
Below find, please, a detailed response to Reviewer critical remarks and comments.
Reviewer comment 2:
The authors must underline the major findings of their work and explain novelty of this study comparatively with their published papers or other similar studies.
Response:
Thanks for this comment and suggestion. The paper presents the method of production as well as the mechanical and strength properties of the active filling, which ensures high technological effects in anaerobic wastewater treatment processes. The works so far has focused only on presenting the technological efficiency related to the efficiency of pollutant removal and biogas production from wastewater of various origins, in rectors of various designs and operated under variable technological conditions. For the first time, detailed data characterizing the filling in terms of physical properties was presented, which is extremely important from the point of view of its implementation in practice and the assessment of the possibility of its use on a large scale. This approach to the issue fully corresponds to the profile of journal Materials. It is not a journal dealing with technological issues of wastewater treatment processes, but rather concentrates on the material side of solutions. As suggested by the Reviewer, major findings and novelty of this study has been added to the introduction section. Please see modified version of manuscript.
Reviewer comment 3:
The Abstract must be revised. The Abstract section refers to the study findings, methodologies, discussion as well as conclusion.
Response:
The authors are grateful for these remarks and agree with the Reviewer's opinion. The abstract has been corrected and supplemented as suggested in the review. The new version of Abstract is as follows:
„A commonly indicated drawback of anaerobic wastewater treatment is the low effectiveness of phosphorus removal. The possibility to eliminate this disadvantage is the implementation of active fillings that contain admixtures of metals, minerals or other elements contributing to wastewater treatment intensification. The aim of the research was to present the active filling produced via microcellular extrusion technology, and to determine its properties and performance in the anaerobic wastewater treatment. The influence of copper and iron admixtures on the properties of the obtained porous extrudate in terms of its functional properties was examined as well. Barus effect increases with the highest content of the blowing agent in the material from 110±12 to 134±14. The addition of metal powders caused increase the extrudate density. The modification of PVC resulted in the highest porosity, amounting to 47.0±3.2% and caused the tensile strength to decrease by about 50%. The determined values ranged from 211.8±18.3 MPa to 97.1±10.0 MPa. The use of filling in anaerobic rectors promoted COD removal, intensified biogas production and eliminated phosphorus with an efficiency of 64.4% to 90.7%, depending on the type of wastewater and applied technological parameters.”
Reviewer comment 4:
Technical and technological bases… is not very clear… is as Introduction part?
Response:
The authors thank the Reviewer for this critical remark. It was improved in the modified version of the manuscript. From the Introduction section, subchapter: "Technical and technological bases" has been removed, modified and placed in the Discussion. Please check with the revised version of the manuscript.
Reviewer comment 5:
The 2.1. Scope of research. In the experimental design, the direct output variables included: temperature distribution along the extrusion head, ºC; rotational speed of the screw, υ s-1;…… The enumeration is hard to watch.
Response:
The authors agree with the Reviewer's opinion. Information presented in subchapter 2.1. Scope of research has been presented in a more readable form. Please see revised manuscript.
Reviewer comment 6:
Please verify all figures.
Response:
All figures have been verified in line with Reviewer suggestion.
Reviewer comment 7:
Please provide comparison with other studies. It is necessary, in tabular form, to provide the comparison studies.
Response:
Authors are grateful for this remark. In the chapter Discussion, a part was introduced to show the technological efficiency of other fillings used in anaerobic reactors for wastewater treatment:
“Other authors used different kinds of fillings in anaerobic wastewater tretament technology. Huang et al. (2020) tested an anaerobic side-stream reactor (ASSR) filled with polyethylene carriers (15% fill) coupled with a membrane bioreactor. COD removal efficiency was 95.4% and a reduction in TP concentration of 40.2% [1] . Gannoun et al. (2008) during the anaerobic purification of cheese whey wastewater in an upflow anaerobic filter (UAF) reactor filled with Flocor material achieved a COD removal efficiency from 72±5 to 90.2±0.5%, the amount of biogas in the range from 0.47±0.05 to 1.15±0.28 dm3∙d at methane content of 89±0.3 - 89±0.3 dm3CH4/kgCODremoved [2]. Filling the polyurethane foam cubes of the concrete reactor contributed to the achievement of the COD removal efficiency at the level of 96% [3]. The use of the multi-fed anaerobic filter (MFR) by Puñal et al. (1999) allowed for a better biomass decomposition compared to the use of a single feed reactor (SFR) [4]. Performance of anaerobic upflow fixed reactors for biomethanation of high-strength cheese whey using different support material such as charcoal, gravel, brick pieces, PVC pieces and pumice stones at 37°C has been studied. Among them the charcoal fixed film reactor showed the best performance [5]. The results are summarized in Table 4. In anaerobic filters equipped with various types of fillings, the efficiency of phosphorus compounds removal was not observed or determined. The main task of the filling was to create a surface for the growing of anaerobic bacteria population and to limit their spread outside the reactor during wastewater treatment. The presented active filling, due to the foreignness of micropores, large active surface and the ability to bind phosphorus, is a competitive solution in relation to the currently used methods.”
The discussed results are summarized in Table 4. Technological efficiency of other fillings used in the processes of anaerobic wastewater treatment. Please see modified version of manuscript.
References:
Huang, J.; Zhou, Z.; Zheng, Y.; Sun, X.; Yu, S.; Zhao, X.; Yang, A.; Wu, C.; Wang, Z. Biological nutrient removal in the anaerobic side-stream reactor coupled membrane bioreactors for sludge reduction. Bioresour Technol 2020, 295, 122241. https://doi.org/10.1016/j.biortech.2019.122241.
Gannoun, H.; Khelifi, E.; Bouallagui, H.; Touhami, Y.; Hamdi, M. Ecological clarification of cheese whey prior to anaerobic digestion in upflow anaerobic filter. Bioresour Technol 2008, 99(14), 6105–6111. https://doi.org/10.1016/j.biortech.2007.12.037.
Ratusznei, S.M.; Rodrigues, J.A.D.; Zaiat, M. Operating feasibility of anaerobic whey treatment in a stirred sequencing batch reactor containing immobilized biomass. Water Sci. Technol. 2003, 48, 179–186.
Puñal, A.; Méndez-Pampín, R.J.; Lema, J.M. Characterization and comparison of biomasses from single- and multi-fed upflow anaerobic filters. Bioresour. Technol. 1999, 68, 293–300. https://doi.org/10.1016/S0960-8524(98)00147-3.
Patel, P.; Desai, M.; Madamwar, D. Biomethanation of cheese whey using anaerobic upflow fixed film reactor. J. Ferment. Bioeng. 1995, 79, 398–399. https://doi.org/10.1016/0922-338X(95)94006-D.
Reviewer comment 8:
The Conclusion must be revised. The main results must be presented in this section.
Response:
The authors agree with the Reviewer's opinion. Conclusions have been supplemented with the most important results. Modified version of the conclusions below:
“A growing interest has been observed in the last two decades in anaerobic technologies which are perceived as competitive to conventional aerobic methods. Unfortunately, their drawback is the low effectiveness of removing biogenic compounds, including mainly phosphorus, which is perceived as the most important factor responsible for the eutrophication processes of natural aquatic ecosystems. This manuscript presents the production process, characteristics and technological performance of an active filling made in the process of microcellular extrusion with the addition of metal powders.
Filling modification with the blowing agent has been shown to modify the structure of the extrudate from solid to porous only in the inner layer of the filling. On the other hand, the addition of copper and iron powders to the filling modified the outer surface of the pipe, but still more metal powders settled on the inner surface of the extrudate. It was found that the Barus effect and porosity increased with the increasing content of the blowing agent in the material, while the addition of metal powders caused no changes in their values. Barus effect increases from 110±12 to 134±14, porosity whereas amounting to 47.0±3.2% and caused the tensile strength to decrease by about 50%. Filling modification with the blowing agent decreased the extrudate density and tensile strength by about 50%. A greater decrease in tensile strength was observed with the addition of copper powder to 97.1±10.0 MPa, compared to PVC modification using iron powder. The use of filling in anaerobic rectors promoted COD removal, intensified biogas production and eliminated phosphorus with an efficiency of 64.4% to 90.7%, depending on the type of wastewater and applied technological parameters.”
Reviewer comment 9:
Please present full form of some. abbreviations when they first appeared. For example Afs, row 71.
Response:
The authors are grateful to the Reviewer for this suggestion. All abbreviations and acronyms used in the manuscript have been checked and explained.
Reviewer comment 10:
The references aren’t in journal’s format. Please respect guide for authors. For example: Ashraf, A.; Ramamurthy, R.; Rene, E. R. Wastewater treatment and resource recovery technologies in the brewery industry: 449 Current trends and emerging practices. Sustainable Energy Technologies and Assessments 2021, 47, 450 101432. Journal wasn’t abbreviated. There are journals with or without abbreviation, etc.
Response:
The authors thank the Reviewer for finding and pointing out these mistakes. Everything has been thoroughly checked and corrected in accordance with the requirements of Materials journal.
Reviewer comment 11:
If the paper is in Materials journal topics, please proved 2 references from this journal (last year).
Response:
This has been done. The following references were used in the manuscript:
Cepan, C.; Segneanu, A.-E.; Grad, O.; Mihailescu, M.; Cepan, M.; Grozescu, I. Assessment of the Different Type of Materials Used for Removing Phosphorus from Wastewater. Materials 2021, 14, 4371. https://doi.org/10.3390/ma14164371
Limousy, L.; Thiebault, T.; Brendle, J. New Materials and Technologies for Wastewater Treatment. Materials 2022, 15, 1927. https://doi.org/10.3390/ma15051927.
Round 2
Reviewer 2 Report
The comments have been addressed.
Reviewer 3 Report
The manuscript was improved in accord with recommendations.